# New Potential Biological Limiters of the Main Esca-Associated Fungi in Grapevine

**DOI:** 10.3390/microorganisms11082099

**Published:** 2023-08-17

**Authors:** Francesco Mannerucci, Giovanni D’Ambrosio, Nicola Regina, Domenico Schiavone, Giovanni Luigi Bruno

**Affiliations:** Department of Soil, Plant and Food Sciences, University of Bari Aldo Moro, Via Amendola 165/A, 70126 Bari, Italy; francesco.mannerucci@uniba.it (F.M.); domenico.schiavone1@uniba.it (D.S.)

**Keywords:** *Aphanocladium album*, *Pleurotus ostreatus*, *Pleurotus eryngii*, *Trichoderma harzianum*, antagonist index, hyphal interaction, deadlock, inhibition, re-isolation, biocontrol

## Abstract

The strains *Trichoderma harzianum* TH07.1-NC (TH), *Aphanocladium album* MX95 (AA), *Pleurotus eryngii* AL142PE (PE) and *Pleurotus ostreatus* ALPO (PO) were tested as biological limiters against *Fomitiporia mediterranea* Fme22.12 (FM), *Phaeoacremonium minimum* Pm22.53 (PM) and *Phaeomoniella chlamydospora* Pc22.65 (PC). Pathogens were obtained from naturally Esca-affected ‘Nero di Troia’ vines cropped in Grumo Appula (Puglia region, Southern Italy). The antagonistic activity of each challenge organism was verified in a dual culture. TH and PO completely overgrew the three pathogens. Partial replacement characterized PE-FM, PE-PM, PE-PC and AA-PC interactions. Deadlock at mycelial contact was observed in AA-FM and AA-PM cultures. The calculated antagonism index (AI) indicated TH and PE as moderately active antagonists (10 < AI < 15), while AA and PO were weakly active (AI < 10). The maximum value of the re-isolation index (s) was associated with deadlock among AA-PM, AA-PC and PE-FM dual cultures. The tested biological limiters were always re-isolated when PO and TH completely replaced the three tested pathogens. TH and AA confirmed their efficiencies as biological limiters when inoculated on detached canes of ‘Nero di Troia’ in dual combination with FM, PC and PM. Nevertheless, additional experiments should be performed for a solid conclusion, along with validation experiments in the field.

## 1. Introduction

The disease known as ‘Esca’ is one of the longest-recognized and most devastating threats to grapevines (*Vitis vinifera*) and viticulture around the world [1]. The most recent studies consider this mycopathy as different diseases that overlap in the same vine or develop at different stages of plant life [1,2,3,4,5]. Cross-sections of rooted cuttings, trunk, branches, and shoots show brown to black spots often accompanied by a dark, viscous exudate (‘black goo’). Longitudinally, xylem necrosis extends in columnar strips, called ‘brown wood streaking’ [1,2,3,4,5]. Brown wood streaking affects rooted cuttings (also named brown wood streaking of grapevine cuttings) and young (2–7 years) vines (defined as Petri disease, formerly also known as black goo, slow dieback, and Phaeoacremonium grapevine decline) [1,2,3]. Inside the trunk or the branches, the wood shows ‘white rot’ (which is the origin of the name Esca), as well as cracking or fissuring of the bark and wood (known as ‘mal dello spacco’—cracking disease—in Italy) [1,3,6]. On leaves, small pale green or chlorotic zones, roundish or irregular, occur dispersed to the veins or adjacent the margin. Progressively, these areas increase, merge, and, in part, necrotize, and at last leave only a narrow strip of unaffected green tissue along the main veins. Depending on the cultivar, dead tissues appear dark brown to red-brown and the diseased leaves develop a ‘tiger stripes’ appearance. Occasionally, the necrotic zones of the lamina desiccate and detach and leaf margins become irregular [7]. These symptoms characterize grapevine leaf stripe disease, previously named ‘young Esca’ [1,2,3,4]. Brown wood streaking of rooted cuttings, Petri disease, and grapevine leaf stripe disease manifest in newly grafted plant material, young vines, and adult plants, respectively [1,3]. The concomitant presence of grapevine leaf stripe disease and white rot is described as ‘Esca proper’. Berries display tiny brown spotting (‘black measles’), shriveling, and wilting [1,2,3,4]. Within the Esca complex, white rot and Esca proper can also show apoplectic symptoms: the sudden wilting of whole vines or individual vine arms within a few days, with leaves and grape clusters that usually remain attached to the plant [1,5].

Members of the basidiomycetous genus *Fomitiporia* (*F. mediterranea* in Europe and the Mediterranean area) are mainly associated with white rot [8]. *Phaeomoniella chlamydospora* and *Phaeoacremonium minimum* (syn. *P. aleophilum*) are the most important etiological agents of brown wood streaking in cuttings, Petri disease, and grapevine leaf stripe disease [1,3,5,7,8,9,10]. No pathogens have been associated with symptomatic leaves or berries of diseased plants. Symptoms on leaves and berries are thought to be related to vine susceptibility and oldness, the implicated pathogens, and pedoclimatic and physiological features [2,5,9,11]. From pycnidia, perithecia or basidiomata embedded in the bark of infected grapevines, and/or from other infected woody hosts surrounding vineyards, produce conidia, ascospores, or basidiospores produced by Esca-associated pathogens spread by rain droplets and wind, land on receptive pruning wounds, and initiate infection [1,5,12]. Infected propagating material also causes disease spread in the field [7,12].

Despite the amount of research devoted to the Esca complex, the frequency of grapes showing Esca-related symptoms has increased worldwide [11]. Thus far, no curative methods are available. In the past, arsenite (revoked because of carcinogens in humans and toxicity towards the environment) spray application alleviated leaf symptoms’ expression [13]. Different strategies are applied, both in nurseries and fields, to limit the occurrence of ‘Esca’ symptoms [1]. Preventive practices, e.g., pruning wound protection and infected stock elimination, are recommended to reduce Esca complex spread [1,4,14] but are insufficient to guarantee effective control [13,14,15,16]. On the other hand, invasive methods (e.g., trunk renewal or surgery, regrafting, and dry rot removal) mitigate the loss of productivity over the years [1,12,13,14,15,16]. No efficient chemical control is yet available. Trunk applications of fosetyl-Al provide reliable results in the reduction of browning and leaf symptom manifestation [13,17]. In vitro treatments with carbendazim, flusilazole, or tebuconazole induced 50% inhibition of *P. chlamydospora* and *P. minimum* mycelia growth, while azoxystrobin, carbendazim, and thiram reduced conidia germination [18]. Hot water (50 °C) treatment for 30 min reduced the *P. chlamydospora* load in propagation material [12]. Different concentrations of cysteine, FeSO_4_, salicylic acid, and fosetyl-Al, alone or in combination, showed antifungal activity against *P. chlamydospora* and *P. minimum*, as well as systemic action and bioactive stimulation on vines [14]. Preventive pruning wound protection seems the most effective strategy [1,12].

Several studies place in biological control the possibility of counteracting the deleterious effects of Esca complex pathogens [1,15,16,17,18,19,20,21,22]. Promising is the application of biocontrol agents, including bacteria, oomycetes, and fungi. Favorable are bacteria of the genera *Acinetobacter*, *Bacillus*, *Brevibacillus*, *Curtobacterium*, *Enterobacter*, and *Paenibacillus*, the oomycote *Pythium oligandrum*, and mycoparasite fungi of the genus *Trichoderma*, still with innovative, eco-friendly hybrid nanomaterials [1,15,19,20].

Strains of *Bacillus subtilis* have revealed in vitro and in planta antagonistic traits against grapevine trunk pathogens, including Esca-associated fungi, both in pruning wound protection and in nurseries [21]. *Bacillus subtilis* and *Bacillus amyloliquefaciens* interact with *P. chlamydospora* and *P. minimum* hyphae via antibiosis [15,22,23]. *Bacillus pumilus* (S32) and *Paenibacillus* isolates S18 and S19 exhibit action by producing volatile compounds and diffusible antibiotic substances and inducing grapevine defense-related gene expression [15,24]. *Enterobacter* isolate S24, *Bacillus reuszeri* strains S28, S31, and S27, *Bacillus* isolate S34, *Pantoea illinoisensis* strain S13, *Pantoea agglomerans* strains S1 and S3, and *Bacillus firmus* strain S41 are efficient biological control agents for Esca-associated fungi on rooted cuttings under greenhouse conditions [15,24]. *Pseudomonas protegens* strain MP12, obtained from a forest soil sample, and *Pseudomonas protegens* strain DSM 19095T significantly inhibited *P. chlamydospora* and *P. minimum* in vitro growth [15,25]. The volatile compound produced by *Pseudomonas* isolate S45, *Stenotrophomonas* isolate S180, and *Novosphingobium* isolate S112 induces deleterious effects on *F. mediterranea* mycelia, while *Enterobacter* isolate S11, *Paenibacillus* isolates S150 and S270, *Weeksellaceae* S259, and *Bacillus* strain S5 even promoted *F. mediterranea* growth in a dual culture [26]. Strains of the cellulolytic and xylanolytic *Paenibacillus* sp. display a synergistic interaction with *F. mediterranea* and enhance wood degradation [26].

Strains of *Actinobacteria* and *Streptomyces* isolates E1 and R4 induced a significant reduction in infection rates at the lower end of the rootstock in the context of Petri disease [15,27,28].

*Streptomyces plymuthica*, *Bacillus velezensis*, *Pseudomonas chlororaphis*, and isolates of *Micromonospora* sp. strongly inhibited *P. minimum* in vitro growth [15,27,29].

The application of *Pythium oligandrum* at the root level reduced *P. chlamydospora* and *P. minimum* necroses in the stem and triggered the plant defense pathways, including PR proteins, phenylpropanoid, oxylipin, and oxidoreduction systems [15,30,31].

*Epicoccum layuense*, an ascomycete often associated with the mycobiomes of grapevines, through in vitro dual culture, inhibited the growth of *P. chlamydospora*, *F. mediterranea*, and some *P. minimum* strains without colony contact, suggesting the production of inhibitory compounds [32]. *E. layuense* strain E24 colonized the rooted grapevine cuttings of Cabernet Sauvignon and Touriga Nacional cultivars under greenhouse conditions and decreased (31–82%) the wood symptomatology via chemical interaction and competition for space, depending on the pathogen and grapevine cultivar [32]. *Epicoccum mezzettii* E17 overgrew *F. mediterranea* in vitro and competed for space and nutrients [32].

Encouraging results regarding the management of Esca-associated fungi have been obtained with the application of Remedier^®^ (ISAGRO S.p.A., Milan, Italy) [1,15,33] and Ecofox Life^®^ (ISAGRO S.p.A., Milan, Italy), bio-fungicides containing *Trichoderma asperellum* and *T. gamsii*. Strains of different species of *Trichoderma*, including *T. atroviride*, *T. harzianum*, *T. hamatum*, *T. longibrachiatum*, and *T. gamsii*, effectively controlled *P. chlamydospora* and *P. minimum* in vitro and under greenhouse, field, and nursery conditions, [1,15,22,34]. Hyphae of *T. atroviride* ATCC74058 and *T. harzianum* ATCC 26799 overgrew *P. chlamydospora* and *P. minimum*, competed for nutrients, utilizing carbon and nitrogen sources, induced direct antagonism, and allowed a 90% growth reduction [15,35]. Commercial strains of *T. harzianum* and *T. atroviride* overgrew *P. chlamydospora* [15,22,35]. *T. atroviride* strains USPP-T1 and USPP-T2 stopped *P. minimum* growth by coiling or disintegrating hyphae [35]. Strains of different *Trichoderma* spp. protected grapevine pruning wounds in nurseries and reduced the incidence of *P. chlamydospora* and *P. minimum* after inoculation under field conditions [22,28]. The application of *T. harzianum* (Trichodex^®^, Fertiberia, Sevilla, España) at rooting under organic nursery conditions reduced *P. chlamydospora* infection [36,37,38]. *T. harzianum* T39 (Trichodex^®^) and *T. longibrachiatum* treatments on cuttings reduced the necrosis length, caused by *P. chlamydospora* inoculation at the rootstock via the enhancement of the grapevine defenses [15,36,37,38]. A *Trichoderma koningii* strain TK7 suspension’s dip application on roots reduced the incidence of *P. chlamydospora* infection in the field on a young grafted Spanish Tempranillo cultivar [15,28].

Strains of *Clonostachys rosea*, in vitro, overgrew and inhibited *P. chlamydospora* and *P. minimum* through antibiosis and mycoparasitism [15,39]. Under greenhouse conditions, amended soil with the endophytic *C. rosea* strain 19B/1 significantly decreased the length of the necrotic lesions caused by *P. chlamydospora* [39]. *Lecanicillium lecanii* (ATCC 46578) reduced in vitro the growth of *P. chlamydospora* and *P. minimum* by carbon and nitrogen competition [34] or by the production of antibiotic compounds [39]. *Fusarium oxysporum* strain F2 reduced *P. chlamydospora* in vitro growth by 43% and sporulation by 90%; nonetheless, the strain showed no reduction in the discoloration length inside the trunk, despite an 82% reduction in the *P. chlamydospora* DNA amount [40].

This work aimed to assess the antagonistic activity of four new biological limiters against the three most important Esca-associated fungi of grapevine. The challenge organisms applied included *Trichoderma harzianum*, the chitinolytic fungus *Aphanocladium album*, and two xylotrophic fungi belonging to the genus *Pleurotus*. Radial growth inhibition, the morphology of the interaction, and the antagonism index were used to evaluate the in vitro dual interactions. The number of positive re-isolations after pathogen and biological limiter interaction was calculated. *T. harzianum* and *A. album* were additionally evaluated on *V. vinifera* cv Nero di Troia detached canes against the three main fungal species associated with the Esca complex.

## 2. Materials and Methods

### 2.1. Strains, Media, and Growth Conditions

Target organisms (Table 1) were isolated from wood samples of 12-year-old cv Nero di Troia vines, grafted on 157-11 located in the countryside of Grumo Appula (Puglia region, southern Italy) and trained to a ‘tendone’ not irrigated.

In the 2020 and 2021 growing seasons, the vineyard was surveyed, and vines with typical Esca symptoms were selected (Figure 1). During the winter pruning operations (January 2022), samples of branches were taken from five symptomatic vines. Internal symptoms through the wood, revealed after cutting, included vascular streaking, light brown to black discoloration, and white rot (Figure 1). Wood samples were wetted with denatured ethanol and surface-sterilized by a flame. From each sample, 15 wood chips (2 mm × 2 mm × 2 mm) were aseptically cut from the margins of diseased wood and plated (5 chips per dish) either on potato dextrose agar (PDA; from Oxoid Part of Thermo Fisher Scientific—Microbiology, Hampshire, UK) amended with 100 mgL^−1^ of each streptomycin and ampicillin (PDA-SA) or PDA modified by adding 1 mL L^−1^ of thiabendazole lactate (PDA-T: 2.3 g of thiabendazole in 10 mL of lactic acid). The incubation of Petri dishes occurred at 24 ± 1 °C in the dark for up to 28 days. Emerging colonies were hyphal-tip-purified. Isolates were identified based on their microscopic morphological characteristics. Strains CBS 229.95, CBS 631.94, and DBPV-1 of *P. chlamydospora*, *P. minimum*, and *F. mediterranea*, respectively [41], were used as a reference for morphological features. The obtained strains were stored on PDA slants in Wheaton bottles at 4 ± 1 °C in the fungal collection of the Department of Soil, Plant and Food Sciences (Di.S.S.P.A.), Plant Pathology Section, University of Bari. 

*Aphanocladium album* MX95 (Patent N° 00041374382), *Pleurotus ostreatus* ALPO, *Pleurotus eryngii* AL142, and *Trichoderma harzianum* TH07.1-NC strains (Table 1), available at the fungal collection of the Di.S.S.P.A. Plant Pathology Section, were revitalized on PDA at 25 ± 1 °C in the dark. All fungi were routinely grown on PDA at 25 ± 1 °C in the dark.

### 2.2. Growth Rate

Potential biological limiters and target organisms were singly grown in a 90 mm Petri dish containing 18 mL of PDA medium. A plug (3 mm in diameter) of each strain, collected from the margins of actively grown cultures, was placed 1 cm from the border of the plate on the line of the dish diameter. Inoculated plates were sealed with Parafilm M and incubated in darkness at 25 ± 1 °C. Radius measurements were performed every eight hours following the line of the dish diameter. The average of the daily radius increment was calculated. All strains were tested in triplicate and the experiment was repeated at least two times.

### 2.3. In Vitro Dual Culture Interactions

Dual interactions between the tested strains (Table 1) were performed in plastic Petri dishes (diameter 90 mm, height 15 mm) containing PDA (18 mL per plate). Mycelium plugs (3 mm in diameter) were cut from the edge side of the actively growing pure culture and used as an inoculant. The target organism and potential biological limiter plugs were placed together on the same plate on opposite sides, 1 cm from the border of the dish (Figure 2a). As a control, the target organism and potential biological limiter plugs were placed alone. Inoculate plates were sealed with Parafilm M and incubated at 25 ± 1 °C, in the dark.

Based on the differences in the growth rates of each fungal strain, the slower PC was placed 8 days in advance of the faster-growing TH.

Three replicates were maintained for each treatment. The experiment was conducted twice for reproducibility. Plates were observed every eight hours to record the time of the first contact between the two mycelia.

Every day, the colony radii of the target organism were assessed in dual culture and on the control plate. In plates with dual culture, radii were measured in the direction of the potential biological limiter. The measurements evaluated 10 days after inoculation were used to determine the percentage of radial growth inhibition as 100 × [(C − T)/C], where C and T are the mean radii (mm) of the pathogen in the absence of a biological limiter (control) and dual culture, respectively. Based on the radial growth inhibition rates, antagonist activity was considered [42] very high (>75%), high (61–75%), moderate (51–60%), or low (<50%).

Mycelial interactions in dual culture were examined daily and scored with the rating proposed by Badalyan et al. [43,44]. The antagonism index, which defines the ability of a fungus to compete with another species, was calculated as ∑ (n × i), where n is the number of each type or subtype of interaction and i is the corresponding score [43,44]. Biological limiters were considered active (antagonism index > 15), moderately active (antagonism index = 10–15), or weak (antagonism index < 10) antagonists [43,44].

Twenty days after mycelia interaction, the observed results of the competition were quantified after fungal re-isolation. From each plate, seven agar discs (5 mm in diameter) were cut (Figure 2a): five from the interaction zone and two (one per strain) from areas with the presumed growth of only one individual fungus (positions labeled “L” and “P” in Figure 2a). Discs were placed on PDA plates and incubated at 25 ± 1 °C in the dark. The score of the re-isolation success of potential biocontrol agents against each pathogen was quantified as ln[(A + 1)/(P + 1)], where A and P are the mean numbers of the successfully re-isolated biological limiter and target organism, respectively (n = max 5). The estimated value of s ranges between +1.79 and −1.79. Higher values of s indicate greater re-isolation success [45,46].

### 2.4. Interaction on Detached Grapevine Canes

From 20 healthy ‘Nero di Troia’ dormant vines, canes (diameter 10–12 mm) were collected in January 2023. In the laboratory, the canes were cut into one node segment (80 mm length) each 2 cm above the bud to simulate a fresh pruning wound. Cane segments were autoclaved (121 °C, 40 min) and distributed, respecting the polarity, in sterile 50 mL Falcon-type plastic tubes (one per tube) containing 20 mL of sterile distilled water.

For AA, TH, PM, and PC, conidia suspensions were prepared in 0.2% agar water as a bio-adhesive. For FM, a mycelia homogenate was obtained in 0.2% agar water using a Sorvall model 17105 Omni-Mixer homogenizer (DuPont de Nemours, Inc., Wilmington, DE, USA). The viability of conidia and mycelia fragments was assessed on PDA by the decimal dilutions method, adjusted to 10^6^ colony-forming units per milliliter, and used (10 μ) to inoculate the apical end of each cane. AA, TH, PC, PM, and FM were assayed singly and in dual pathogen–limiter arrangements. In dual combinations, target organisms were inoculated 5 days before or after the biological limiters. Canes treated with 0.2% agar water and untreated canes were used as controls. A total of 20 canes per treatment were used. All tubes were incubated at 25 ± 1 °C, in the dark, for 12 weeks. Re-isolations were performed to verify the vitality of the fungal strains inoculated and their progression inside the cane tissue. Each apical end was gently pressed on the PDASA medium surface to create 5 different imprints. Then, every cane was divided longitudinally. Chips (2 mm × 2 mm × 2 mm) were aseptically taken at approximately 5, 10, 15, and 20 mm (Figure 2b) from the upper end and plated (5 per dish) on PDASA. Inoculated dishes were incubated at 25 ± 1 °C, in the dark, for 28–45 d. The developed fungal colonies were identified based on their macro- and microscopic morphological characters. The percentage of colonization frequency of re-isolated fungi was estimated as 100 × (N × n), where N is the number of colonies of each species developed from each chip, and n is the total number of plated fragments (n = 15).

### 2.5. Statistical Analysis

Plates and tubes were allotted in a randomized design. Normality and homogeneity of variances were verified with Shapiro–Wilk’s test and Levene’s test, respectively. The standard deviations (sd) were calculated for all mean values. The experimental data obtained were compared using an analysis of variance (ANOVA), followed by the Fisher least significant difference or Kruskal–Wallis test (*p* = 0.05). The data of inhibition radial growth were analyzed as radius values and expressed as a percentage. The frequency of re-isolated fungi percentage was transformed to arcsine before analysis.

## 3. Results

### 3.1. Growth Rate

All the tested strains developed a different growth rate (Figure 3).

TH colonized the entire Petri dish over 72 h (30.0 mm d^−1^ radius increment). PO (6.1 mm d^−1^), PE (4.7 mm d^−1^), and AA (3.7 mm d^−1^) reached, after 96 h, 15, 19, and 24 mm, respectively (Figure 3a). Among the tested pathogens (Figure 3b), PC (5.7 mm d^−1^) was the slowest, while the fastest growth rate was recorded for PM (17.7 mm d^−1^) and FM (18.7 mm d^−1^).

### 3.2. In Vitro Dual Culture Interactions

In the dual cultures, pathogens and potential biological limiters required 56 to 96 h for the first colony contact (Table 2). Short times were associated with organisms showing fast mycelial growth (e.g., TH against FM, PM, or PC). On the contrary, long times affected the interactions between slow-growth organisms (e.g., AA against PM and PC).

The percentage of radial growth inhibition (Figure 4) ranged from 48 to 87% in the PO-FM and TH-PC8 interactions, respectively.

Very high radial growth inhibition was calculated in the AA-PC, PO-PC, PE-FM, PE-PC, PE-PM, TH-PM, TH-PC, and PE-PC8 interactions. High antagonist activity was recovered in the AA-PM and PO-PM relations. Moderate antagonist activity (51–60%) was recovered in the AA-PM and PO-PC groups, while low inhibition rates (about 49%) were retrieved in the PO-FM and AA-FM dual cultures.

The dual culture assays displayed a miscellaneous pattern of mycelial interaction (Figure 5). Following the definition of mycelial interaction proposed by Badalyan et al. [43,44], deadlock (mutual inhibition in which neither strain was able to overgrow the other) at mycelial contact, replacement (overgrowth without initial deadlock), and partial replacement after an initial deadlock with mycelial contact occurred among the pairings tested.

AA showed deadlock at mycelial contact (interaction type A by Badalyan et al. [43,44]) during the interactions with FM and PM. Partial replacement after initial deadlock with mycelial contact (interaction type C_A1_ as suggested by Badalyan et al. [43,44]) was exhibited during AA-PC dual cultures and PE towards FM, PC, and PM. PO and TH completely replaced (interaction type C as indicated by Badalyan et al. [43,44]) the three tested phytopathogenic organisms, including PC inoculated 8 days in advance of TH. Among the 78 pairings tested, replacement of the pathogen by the potential biological limiter was more frequent (53.8%) than partial replacement after initial deadlock with mycelial contact (30.8%) and deadlock (15.4%).

Based on the calculated antagonism index values, AA and PO were weak antagonists, while PE and TH were moderately active antagonists, reaching antagonism indexes of 5.5, 9.0, 10.5, and 12.0, respectively.

The outcomes of re-isolation success in the dual cultures are shown in Table 3. No target pathogens were re-isolated after the interactions PO-FM, PO-PM, PO-PC, TH-FM, TH-PM, TH-PC, and TH-PC8. In all the other combinations, the pathogen was always re-isolated from the agar disc cut at the P position (Figure 2a). FM was re-isolated from the connective line with AA, while PM and PC were obtained from the deadlock lines with PO.

### 3.3. Interaction on Detached Grapevine Canes

Abundant mycelial efflorescence appeared on the cutting surfaces of the detached ‘Nero di Troia’ canes inoculated with AA, FM, FM5-AA, FM5-TH, AA5-FM, AA5-PM, AA5-PC, and TH5-PC (Figure 6).

From the single inoculations with AA, TH, PM, PC, or FM, the inoculated fungi were recovered from all the imprints (Table 4). In dual inoculations AA5-FM, AA5-PM, AA5-PC, FM5-AA, PM5-AA, and PC5-AA, only AA was re-isolated, whereas TH was obtained from the TH5-FM, TH5-PM, TH5-PC, FM5-TH, PM5-TH, and PC5-TH combinations.

A similar pattern was observed from the re-isolations performed with the fragments cut at 5 mm below the upper end of the cutting surface (Table 4).

Negative was the isolation from the longitudinal sections of the canes performed at approximately 10, 15, and 20 mm below the upper end of the cutting surface for all the tested detached canes (Table 4). No organisms were obtained from the re-isolations performed with the imprints and the chips cut at 5, 10, 15, and 20 mm below the apical end of the detached cane used as a non-inoculated control (CN) and the control inoculated with the adhesive suspension WA (Table 4).

## 4. Discussion

The use of microbial antagonists to manage disease agents is one of the current frontiers in reducing the worrying dependence on pesticides, ensuring food safety and security, and protecting the environment and consumers [47,48,49].

*T. harzianum* TH1, *A. album* MX95, *P. ostreatus* ALPO, and *P. eryngii* AL142PE were tested as potential biological limiters against *F. mediterranea*, *P. minimum*, and *P. chlamydospora*, the main pathogens worldwide associated with grapevine Esca complex.

The isolations here carried out from ‘Nero di Troia’ vines confirmed the involvement of *P. minimum* and *P. chlamydospora* in brown wood streaking and *F. mediterranea* in white rot, while their mixture in the wood led to foliar symptoms [1,2,3,4,5,6,7,8,9,10,50].

A dual culture evaluates the antagonistic properties of microorganisms. This technique shows rapidly and clearly the mutual effects of the paired organisms and their interactions. However, the method excludes the host plant and cannot be used for biocontrol agents that allow disease control through tissue colonization, systemic resistance induction, and/or niche competition [43,44,45,51,52].

The growth rate on the PDA medium differentiates the behavior of tested organisms and could support potential biological limiters and pathogens during dual culture interaction.

Following the suggestions of Bell et al. [53], TH and PO are valid antagonists, because they covered completely the surfaces of the three tested pathogens (type C interaction following the Badalyan et al. [43,44] score rating).

This was supported by the antagonism index and the quantification of the re-isolation success. Low values of the antagonism index indicate a weak response of the strain in terms of inhibition, while high antagonism indices are associated with highly competitive and inhibitory properties in the antagonist [43,44]. The quantification of the re-isolation success of a potential biological control agent against a pathogen is a reliable tool to confirm the success of an antagonist in the interaction with the mycelium of a competing pathogen and represents an important parameter in dual culture assays [45]. The re-isolation of both the antagonist and pathogen from the interaction zone allowed us to assume that the challenge strains were not highly active competitors. In contrast, the re-isolation of only the antagonist from the interaction zone suggested the elimination of the challenger organism and obviously higher success in competition [45]. During the interactions labeled as “type c” (replacement), shown by TH and PO against the three tested pathogens, the biological limiter was always re-isolated from the overlapping zone and the area of pathogen presence (e.g., letter P in Figure 2a). This confirms the antagonistic aptitude of TH and PO and their ability to stop or inhibit the growth of challenged organisms [43,44,53]. Re-isolations of pathogens and biological limiters, even from the same discs, taken from the interface zone after type A and type C_A1_ interactions, were obtained in AA-FM, PE-PM, and PE-PC. In the other combinations, the pathogenic organism was replaced by the biological limiter. The outcomes of these experiments allowed us to assume the weak competitive capacity of FM, PM, and PC, as hypothesized for *Scleroconidioma sphagnicola* [45] and *Eutypella parasitica* [46]. A similar re-isolation result after deadlock at a distance (interaction type B) was reported by Koukol et al. [45], which was not recorded in our study.

The detached cane assay, a modification of the single node cutting technique [54], assesses cane fruitfulness, reduces the time needed to evaluate pruning wound treatments and screenings of vine cultivars for grapevine trunk disease development [55], and allows potential biocontrol activity characterization [56]. Here, we adapted the technique as a model for the study of Esca-associated fungi in wood colonization and the interactions with AA and TH, two probable biological limiters. Due to the results of the study of the interaction among *P. chlamydospora*, *P. minumum*, and *F. mediterranea* performed in vitro and on grapevines cv. Italia and Matilde [57], single interaction pathogens vs. biological limiters were here evaluated. During previous in vitro and in planta interactions, *P. chlamydospora* and *P. minumum* competed for the substratum, albeit not directly challenging: *F. mediterranea* overgrew *P. chlamydospora*, and the interaction of *P. minumum* vs. *F. mediterranea* showed a deadlock at mycelia contact. In triple cultures, *P. minumum* in some way “prevented” *F. mediterranea* from overgrowing *P. chlamydospora* [57]. A similar pattern was seen on the woody tissue of both tested grapevine cultivars [57]. Furthermore, in naturally infected vine material, *F. mediterranea* is commonly associated with decayed wood, while *P. chlamydospora* and *P. minimum* are linked to brown wood streaking [1,3,5,7,8,9,10]. Inside the trunks of Esca-affected vines, rotted tissues are often bordered by a brown line that separates rotted from nondecayed wood [2]. It is probable that when colonizing a vacant resource, such as a fresh wound, each invading fungus became competitive and could exclude other micro-organisms. Another theory of wood colonization during Esca development considers *P. chlamydospora* able to reduce plant resistance due to its toxic activity, *P. minimum* affects cell wall integrity through its enzymatic activity, and *F. mediterranea* enables the complete degradation of wood tissues, resulting in white rot formation [1,2,3,6,15,57].

To avoid a competitive association among *P. chlamydospora*, *P. minumum*, and *F. mediterranea*, single pathogen inoculation was here considered.

The detached cane assay confirmed the ability of PM, PC, and FM to colonize pruning wounds and to use this as a penetration method to colonize the wood tissues of the vine [1,2,3,49,50,57]. The experiments showed that the pathogens and potential biological limiters were effective in colonizing the apical end of the cane and progressing inside the woody tissue when singly inoculated. TH and AA also effectively protected pruning wounds against FM, PM, and PC for at least 3 months after treatment and confirmed the antagonistic effect exhibited in the dual cultures.

The four species used in the present study are well-known antagonists of plant-pathogenic fungi and nematodes.

*A. album* (*Fungi*, *Ascomycota*, *Pezizomycotina*, *Sordariomycetes*, *Hypocreales*, *Nectriaceae*) is a necrotrophic mycoparasite of *Puccinia coronata*, *Puccinia hordei*, *Puccinia graminis* f. sp. *avenue*, *Puccinia recondita* f. sp. *triticina*, *Golovinomyces* (*Oidium*) *lycopersici*, *Podosphaera* (*Sphaerotheca*) *fusca*, *Pseudopyrenochaeta* (*Pyrenochaeta*) *lycopersici*, *Meloidogyne incognita*, *Meloidogyne javanica*, and soil-borne plant pathogenic fungi [58,59,60,61,62]. The necrotrophic effects of *A. album* are associated with the production of hydrolytic enzymes (e.g., protease, glucanase) and chitinases [58].

*P. eryngii* and *P. ostreatus* (*Fungi*, *Basidiomycota*, *Agaricomycotina*, *Agaricomycetes*, *Agaricales*, *Pleurotaceae*) are xylotrophic mushrooms extensively cultivated around the world and evaluated as biological control agents for sugar beet nematode *Heterodera schachtii* [63]. In a dual culture, *P. ostreatus* strongly inhibited the mycelia growth of cereal-pathogenic fungi such as *Ceratobasidium cereale* (anamorph *Rhizoctonia cerealis*), *Gaeumannomyces tritici* (formerly *Gaeumannomyces graminis* var. *tritici*), *F. culmorum*, and *Bipolaris sorokiniana* [43]. *P. ostreatus* also combats the mycoparasites *Clonostachys rosea*, *T. harzianum*, *Trichoderma pseudokoningii*, and *Tichoderma viride* [44]. Furthermore, in dual culture assays, strains of *P. ostreatus* and *P. eryngii* exhibited strong inhibitory activity against *F. solani*, *F. oxysporum* f. sp. *lycopersici*, *V. dahliae*, *P. nicotianae*, and *S. sclerotiorum* [62].

*Trichoderma* is a genus of fungi from the family *Hypocreaceae*, commonly associated with the rhizosphere [64]. Several species of this genus and their *Hypocrea* teleomorphs are used in agriculture as bio-regulators of plant growth and biocontrol agents for the management of nematodes and plant diseases [64,65,66]. Among the *Trichoderma* species useful as biocontrol agents, *T. harzianum* is widely used in plant disease control [67]. The species is proposed as a complex of species based on secondary metabolite production, the target pathogen, the host range, and the distribution area [66,67,68,69]. Morphological and molecular characterizations of commercial *T. harzianum* biological control products show the presence of nine new species [67]. Different *Trichoderma* species, including *T. atroviride*, *T. harzianum*, *T. asperellum*, *T. gamsii*, and *T. longibrachiatum*, have been widely used against fungi in the Esca complex during grapevine nursery propagation processes and for pruning wound protection [1,12,15,19,20,33,36,37,38]. Different species of *Trichoderma* provide efficacy against fungi in the *Botryosphaeriaceae* and the *Diatrypaceae* associated with grapevine trunk disease [56]. Antibiosis, mycoparasitism, and nutrient and/or space competition are mechanisms of action associated with *Trichoderma*’s antifungal activity [15,19,20,69]. Therefore, effectiveness in pruning wound protection requires the establishment of a biological agent. For this, *Trichoderma*-based commercial products give the greatest protective action several days after application.

In the present study, TH and AA effectively prevented pruning wound colonization by *F. mediterranea*, *P. minimum*, and *P. chlamydospora* mycelia and conidia spread. They effectively also stopped the colonization developed by each pathogen.

*P. chlamydospora*, *P. minimum*, and *F. mediterranea*, the main fungal species involved in the Esca complex of grapevine, were successfully added to the target organisms of these interesting biological limiters.

Among the four species here tested, *T. harzianum* and *A. album* demonstrate a long history as fungal antagonists. Strains of *T. harzianum*, including the commercial one (Trichodex^®^), effectively controlled *P. chlamydospora* and *P. minimum* in vitro and under greenhouse, field, and nursery conditions [1,15,22,34]. During antagonistic interaction, *T. harzianum* explores different modes of action: it competes for nutrients, utilizing carbon and nitrogen sources, induces direct antagonism, overgrows its “hosts”, induces growth reductions [15,22,35], and reduces infection [36,37,38] and the necrosis length on rootstock and nursery material [15,36,37,38]. The strain DiSSPA TH07.1-NC, tested in this study, was a moderately active antagonist (antagonism index of 12.0), showed overgrowth, inhibited radial growth, and quickly interacted with the tested Esca-associated pathogens. A direct effect of this strain on PC, PM, and FM can be assumed from the score of the re-isolation success and the colonization frequency. No target pathogens were re-isolated after the interactions with TH on PDA plates and from dual-inoculated detached ‘Nero di Troia’ canes. Furthermore, the strain DiSSPA TH07.1-NC was able to colonize the apical end of each treated cane, to produce abundant mycelial efflorescence on the cutting surfaces and penetrate the first 5 mm of canes during 30 days.

Strains of the necrotrophic mycoparasite *A. album* inhibit the sporulation and the growth of several powdery mildew and rust agents, soil-borne plant pathogens, and nematodes [59,60,61,62]. This antagonistic efficiency is supported by the production of hydrolytic enzymes such as protease, glucanase, and several chitinases involved in the cell wall degradation of many phytopathogenic fungi [58].

The strain DiSSPA MX95, tested in this study, was a weak antagonist (antagonism index of 5.5), showing deadlock (mutual inhibition in which neither strain was able to overgrow the other) at mycelial contact against FM and PM and partial replacement after an initial deadlock with mycelial contact against PC. Despite its slow growth rate, DiSSPA MX95 inhibits radial growth and greater re-isolation success on PDA plates. During the experiments on detached ‘Nero di Troia’ canes, the strain DiSSPA MX95 was able to colonize the apical end of each treated cane, to produce abundant mycelial efflorescence on the cutting surfaces and penetrate the first 5 mm of canes.

This was the first application of *P. ostreatus* and *P. eryngii* strains against fungal species involved in the Esca complex of grapevine. The strains DiSSPA ALPO and DiSSPA AL142, tested in this study, were moderately active antagonists (antagonism index of 9.0 and 10.5 for PO and PE, respectively), showed overgrowth, inhibited radial growth, and quickly interacted with the tested Esca-associated pathogens. A direct effect of these strains on PC, PM, and FM can be assumed from the score of the re-isolation success: no target pathogens were re-isolated after the interactions with PO and PE on PDA plates.

Further screening of the strains used in this study should be conducted to assess their efficacy against other grapevine trunk disease fungi, including *Ilyonectria* spp., *Cadophora luteo-olivacea*, *Diplodia seriata*, and *Neofusicoccum parvum*, commonly found in nursery-propagated material and responsible for young grapevines’ decline and death.

A shortcoming of our study is related to the slight repetitions performed. Additional experiments, testing a broader range of *P. chlamydospora*, *P. minimum*, and *F. mediterranea* strains, are needed for the confirmation and clarification of our results and a solid conclusion. However, further testing should be performed to optimize the application methods (e.g., dose and time), together with validation experiments in the field. Moreover, combinations of AA, PO, PM, and TH will be tested.

## 5. Conclusions

The interactions between the three most important species isolated from the wood of Esca-complex-affected vines and the four strains tested as biological limiters in the dual cultures, here analyzed in terms of the first contact between the two mycelia, the percentage of radial growth inhibition, the type of mycelial interaction, the antagonism index, and the quantification of fungal re-isolation, suggested that the tested strains *P. ostreatus* ALPO, *P. eryngii* AL142, *A. album* MX-95, and *T. harzianum* TH07.1-NC were competitive and caused the greatest inhibition of the three pathogens. Interactions on detached grapevine canes among TH and AA against FM, PC, and PM confirmed the in vitro effects. In particular, *A. album* MX-95 and *T. harzianum* TH07.1-NC are the most promising strains to be used as biological control agents against the three main pathogens associated with the Esca complex of grapevine.

## Figures and Tables

**Figure 1 microorganisms-11-02099-f001:**
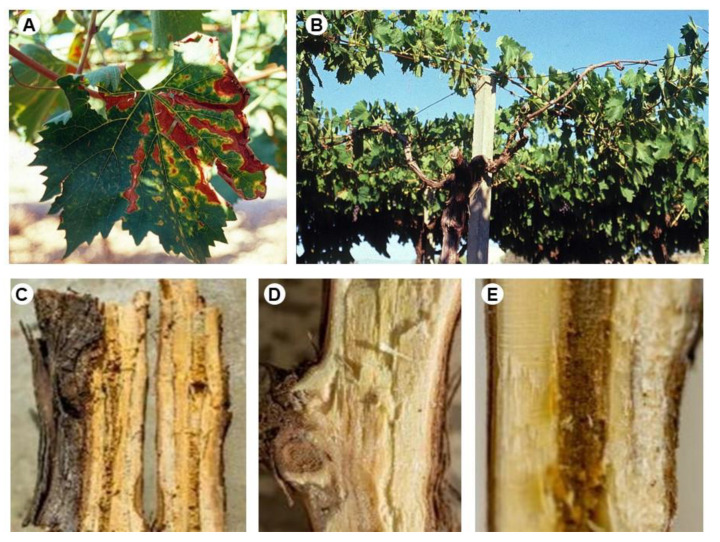
Symptoms on *Vitis vinifera* cv Nero di Troia plants used for isolation experiments: ‘tiger-stripes’ (**A**) and apoplexy (**B**), brown wood streaking (**C**–**E**), and white rot (**C**).

**Figure 2 microorganisms-11-02099-f002:**
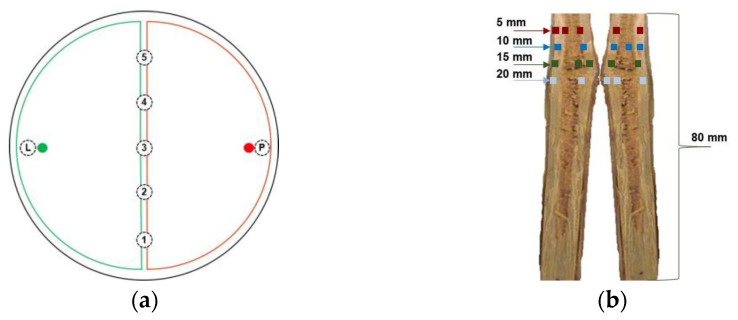
Schemes showing the positions of (**a**) inoculation plugs (● and ●) and discs taken from the interaction zone (1–5), the potential biological limiter (L, ●), and the pathogen (P, ●); (**b**) approximate sites of chip recovery (■, ■,■, ■) for grapevine detached cane re-isolation experiments.

**Figure 3 microorganisms-11-02099-f003:**
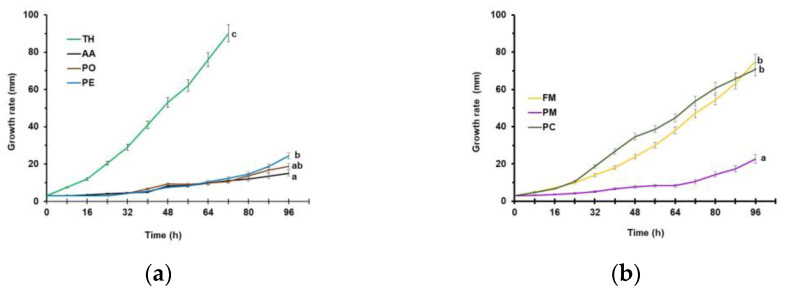
Growth rate at 25 ± 1 °C in the dark on 90 mm Petri dishes containing potato dextrose agar of potential biological limiters (**a**) and target pathogens (**b**). Data are the means of six replicates ± sd. For acronym definitions, see Table 1. Growth curves with different letters are significantly different according to Fisher’s least significant difference test at *p* ≤ 0.05.

**Figure 4 microorganisms-11-02099-f004:**
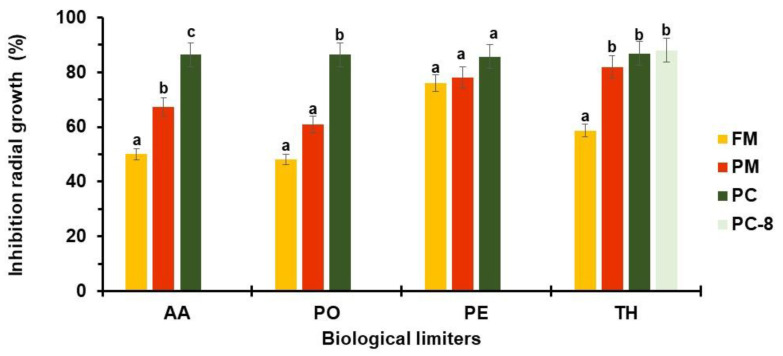
Percentage of inhibition radial growth in dual cultures on potato dextrose agar of AA, PO, PE, and TH against FM (■), PM (■), and PC, inoculated concomitantly (■) or 8 days in advance of TH (■). Data are the means of six replicates ± sd. For each potential biological limiter, values accompanied by the same letters are not significantly different (*p* ≤ 0.05) according to Fisher’s LSD test. For acronym definitions, see Table 1.

**Figure 5 microorganisms-11-02099-f005:**
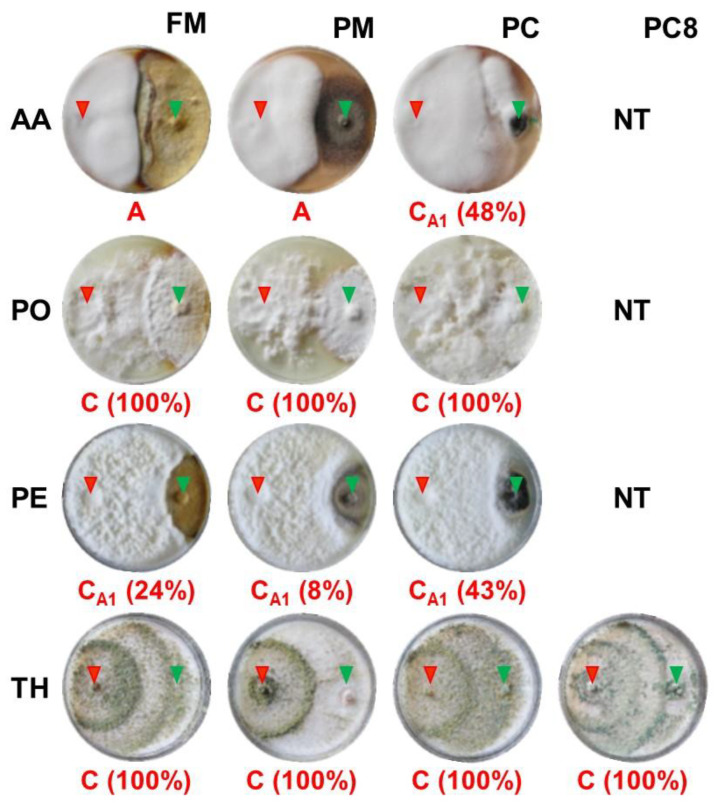
Mycelial interactions after 28 days of dual culture on potato dextrose agar between the potential biological limiters (AA, PO, PE, or TH) and the target pathogens (FM, PM, or PC). PC was inoculated concomitantly (PC) or 8 days in advance of TH (PC8). For acronym definitions, see Table 1. Red letters show the type of interaction as proposed by Badalyan et al. [43,44]: A = deadlock (mutual inhibition in which neither strain was able to overgrow the other) at mycelial contact, C = replacement (overgrowth without initial deadlock), C_A1_ = partial replacement after an initial deadlock with mycelial contact. In brackets is the percentage of overgrowth 20 days after inoculation. NT = not tested. Red and green arrows indicate the sites where biological limiters and pathogens were inoculated, respectively.

**Figure 6 microorganisms-11-02099-f006:**
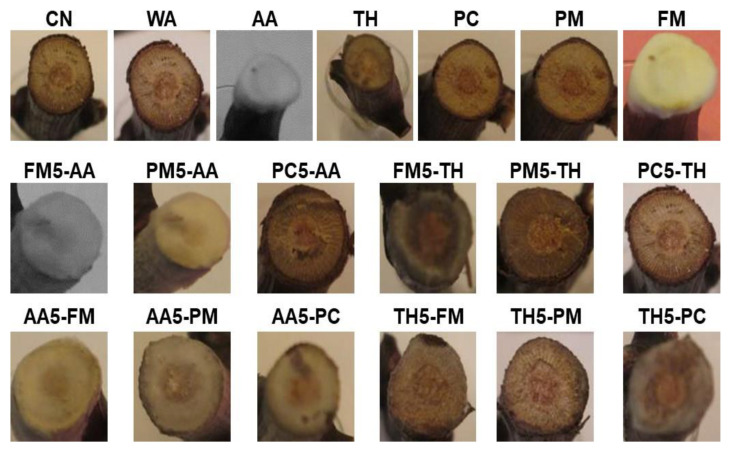
Aspect of detached ‘Nero di Troia’ canes inoculated with 100 µL of 10^6^ mL^−1^ viable conidia (AA, TH, PM, and PC) or mycelium fragment (FM) suspensions in 0.2% agar in water (WA) as the adhesive medium. Strains were inoculated individually (AA, TH, PC, PM, FM) or in dual combination, in which the pathogens were distributed 5 days before the biological limiters (FM5-AA, PM5-AA, PC5-AA, FM5-TH, PM5-TH, PC5-TH) or biological limiters were deposited 5 days before the pathogens (AA5-FM, AA5-PM, AA5-PC, TH5-FM, TH5-PM, TH5-PC). CN = non-inoculated control. Pictures were taken 35 days after the first deposition.

**Table 1 microorganisms-11-02099-t001:** Strains used in this study.

Organism	Code	Acronym
Potential biological limiters		
*Aphanocladium album*	DiSSPA ^1^ MX95	AA
*Pleurotus ostreatus*	DiSSPA ALPO	PO
*Pleurotus eryngii*	DiSSPA AL142	PE
*Trichoderma harzianum*	DiSSPA TH07.1-NC	TH
Target pathogens		
*Fomitiporia* *mediterranea*	DiSSPA Fme22.12	FM
*Phaeoacremonium* *minimum*	DiSSPA Pm22.53	PM
*Phaeomoniella* *chlamydospora*	DiSSPA Pc22.65	PC

^1^ DiSSPA: Department of Soil, Plant and Food Sciences, University of Bari Aldo Moro, Bari, Italy.

**Table 2 microorganisms-11-02099-t002:** Time (hours) required for the first contact between the tested potential biological limiters and phytopathogenic organisms ^1^.

Phytopathogenic	Biological Limiters ^2^			
Organisms	AA	PO	PE	TH
FM	72 ±5.06 a	88 ± 5.06 a	88 ± 5.06 a	56 ± 5.06 a
PM	96 ± 5.06 b	88 ± 5.06 a	88 ± 5.06 a	64 ± 5.06 b
PC ^3^	96 ± 5.06 b	88 ± 7.16 a	88 ± 7.16 a	64 ± 7.16 b
PC8 ^3^	NT ^4^	NT	NT	64 ± 5.06 b

^1^ For acronym definitions, see Table 1. ^2^ Data are the means of six replicates ± sd. Within each column, data with different letters are significantly different according to Fisher’s least significant difference test at *p* ≤ 0.05. ^3^ PC was inoculated concomitantly (PC) or 8 days in advance (PC8) of TH. ^4^ NT = not tested.

**Table 3 microorganisms-11-02099-t003:** Mean scores of re-isolation success of potential biological limiters against each pathogen from in vitro dual cultures ^1,2^.

Biological Limiters	Phytopathogens ^3^			
	FM	PM	PC ^4^	PC8 ^4^
AA	1.09 ± 0 a	1.79 ± 0 b	1.79 ± 0 b	NT ^5^
PO	*	*	*	NT
PE	1.79 ± 0 b	1.09 ± 0 a	1.09 ± 0 a	NT
TH	*	*	*	*

^1^ The score of re-isolation success was quantified as ln[(A + 1)/(P + 1)], where A and P are the mean numbers of the successfully re-isolated biological limiter and target organism, respectively (n = max 5). ^2^ For species acronym definitions, see Table 1. ^3^ The data are the means of six replicates ± sd. Within each row, values with different letters are significantly different according to the Kruskal–Wallis test at *p* ≤ 0.05. ^4^ PC was inoculated concomitantly (PC) or 8 days in advance of TH (PC8). ^5^ NT = not tested. * = re-isolated with only the biological limiter.

**Table 4 microorganisms-11-02099-t004:** Colonization frequency of re-isolated fungi (%) ^1^ from ‘Nero di Troia’ detached canes inoculated with AA, TH, FM, PM, PC, and their dual combination with pathogens deposited 5 days before (FM5-AA, PM5-AA, PC5-AA, FM5-TH, PM5-TH, PC5-TH) or after (AA5-FM, AA5-PM, AA5-PC, TH5-FM, TH5-PM, TH5-PC) the biological limiters ^2^.

Treatments	Strains	Position ^4^				
		Cutting Surface ^3^	Longitudinal Section (mm below the surface)			
			5	10	15	20
CN ^5^		NFI ^7^	NFI	NFI	NFI	NFI
WA ^6^		NFI	NFI	NFI	NFI	NFI
AA	AA	100	100	NFI	NFI	NFI
TH	TH	100	100	NFI	NFI	NFI
FM	FM	100	100	NFI	NFI	NFI
PM	PM	100	100	NFI	NFI	NFI
PC	PC	100	100	NFI	NFI	NFI
FM5-AA	AA FM	100 0	100 0	NFI	NFI	NFI
PM5-AA	AA PM	100 0	100 0	NFI	NFI	NFI
PC5-AA	AA PC	100 0	100 0	NFI	NFI	NFI
FM5-TH	TH FM	100 0	100 0	NFI	NFI	NFI
PM5-TH	TH PM	100 0	100 0	NFI	NFI	NFI
PC5-TH	TH PC	100 0	100 0	NFI	NFI	NFI
AA5-FM	AA FM	100 0	100 0	NFI	NFI	NFI
AA5-PM	AA PM	100 0	100 0	NFI	NFI	NFI
AA5-PC	AA PC	100 0	100 0	NFI	NFI	NFI
TH5-FM	TH FM	100 0	100 0	NFI	NFI	NFI
TH5-PM	TH PM	100 0	100 0	NFI	NFI	NFI
TH5-PC	TH PC	100 0	100 0	NFI	NFI	NFI

^1^ Colonization frequency of re-isolated fungi was estimated as 100 × (N × n), where N is the number of colonies of each species developed from each chip, and n is the total number of plated fragments (n = 15). ^2^ For acronym definitions, see Table 1. ^3^ The data are the means of fifteen imprints per treatment. ^4^ Data are the means of fifteen chips for treatment. ^5^ CN = non-inoculated control. ^6^ WA = 0.2% agar in water. ^7^ NFI = no fungal isolation.

## Data Availability

Data are contained within the article.

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
