# Peer review of "New Potential Biological Limiters of the Main Esca-Associated Fungi in Grapevine"

_microorganisms, 2023, doi:10.3390/microorganisms11082099_

Round 1
Reviewer 1 Report
Overall Summary: Mannerucci et al., test the bio-fungicidal potential of four fungal species against the casual agents of the Esca disease complex affecting grapevines in Italy. They conduct competition assays in-vitro and on detached shoots to test this. Below are some comments to improve the data presentation in the manuscript.
Specific comments:
· Line 133: Should be ‘Inoculated plates’.
· Figure 3.1: This figure should include a statistical comparison for the graphs with significance letter markers.
· Figure 3.1: Are the error bars in the figure standard deviation or standard error based. It is unclear from the explanation why the error bars would have Fisher’s LSD. The statistical test should be denoted with significance letters not error bars.
· Table 3: The datapoints should include error rate using standard error or standard deviation.
· Figure 5: From figure 1 and figure 3, it is unclear where the pathogen biological agent was inoculated on the plate in terms of the side where they were inoculated. This should be clearly indicated on the plate in the image for figure 5.
· Table 4 and table 5: The scoring in terms of what the numerical value means in terms of isolation of a fungus needs to be explained in these tables. It is unclear why the scale varies in table 4 from ~1 to 100 in table 5. Is this a direct numerical value or a percentage. The method used for calculating this should also be mentioned and explained in the results section while explaining the table.
· The study only includes inoculations with individual pathogenic strains in the detached shoot assays. However, due to the condition being a disease complex, multiple strains should be inoculated together to study if the biological agents would be able to control the condition.
· The discussion section restates a lot of the results. Instead, the detailed reporting of the results should be moved into the results section. The conclusions and limitations of the study should be included in the discussion section. (Lines 309-330)
Reviewer 2 Report
The manuscript written by Mannerucci et al. reports some interesting results concerning Trichoderma harzianum, Aphanocladium album, Pleurotus eryngii, and Pleurotus ostreatus as an effective alternative to suppress Fomitiporia mediterranea, Phaeoacremonium minimum, and Phaeomoniella chlamydospora. The study is backed up with experimental data and evidence, which are currently followed for similar types of work worldwide.
In totality, the conceptualization, designing of experiments, and overall write-up are good. However, it needs major corrections and there are some queries which the authors should kindly respond to make it good, to the best of my knowledge.
As a general point, overusing abbreviations could make this article challenging to comprehend. Please consider this while reviewing/editing.
Introduction
The introduction provides insufficient information regarding the reference species' ability to function as biological limiters, given the existence of commercial products based on Trichoderma.
Moreover, Bacillus strains effectively act as biocontrol agents against fungi responsible for grapevine trunk diseases. The Introduction and Discussion sections should provide further information, including references to recent research in the field.
Materials and methods
The fungal pathogens were isolated during this study. Their identification was based only on their microscopic morphological characters. Macroscopic observations could be added to provide a more complete identification process. Were reference strains used for confirmation purposes? Also, sequencing of the 5.8S rDNA gene confirms correct identification at the species level.
Additional information should be included in the Results section to support the identification of fungal pathogenic strains.
Have you used commercial bio-fungicides based on Trichoderma to support your results of tested interaction on detached grapevine canes?
Lines 145 and 264: please check if they refer to Figure 1a
Discussions
It is necessary to discuss the advantages and characteristics of the candidate strains in comparison to the reported strains, particularly in the context of grapevine trunk disease.
Conclusion
The conclusion should be shortened and should be scientifically reviewed and rewritten. Based on your experiments, decide which is the best strain to use as a biological limiter.
Round 2
Reviewer 2 Report
The authors have taken most of the comments into account and have revised the manuscript accordingly. The revised manuscript is acceptable and can be published in its present form.